# Dissipation + Utilization = Self-Organization

**DOI:** 10.3390/e25020229

**Published:** 2023-01-26

**Authors:** Harrison Crecraft

**Affiliations:** GeoEx Analytics, Leesburg, VA 20176, USA; harrison@crecraft.net

**Keywords:** dissipative systems, dissipative structuring, evolution, non-equilibrium thermodynamics, ecosystems, origin of life

## Abstract

This article applies the thermocontextual interpretation (TCI) to open dissipative systems. TCI is a generalization of the conceptual frameworks underlying mechanics and thermodynamics. It defines exergy with respect to the positive-temperature surroundings as a property of state, and it defines the dissipation and utilization of exergy as functional properties of process. The Second Law of thermodynamics states that an isolated system maximizes its entropy (by dissipating and minimizing its exergy). TCI’s Postulate Four generalizes the Second Law for non-isolated systems. A non-isolated system minimizes its exergy, but it can do so either by dissipating exergy or utilizing it. A non-isolated dissipator can utilize exergy either by performing external work on the surroundings or by carrying out the internal work of sustaining other dissipators within a dissipative network. TCI defines a dissipative system’s efficiency by the ratio of exergy utilization to exergy input. TCI’s Postulate Five (MaxEff), introduced here, states that a system maximizes its efficiency to the extent allowed by the system’s kinetics and thermocontextual boundary constraints. Two paths of increasing efficiency lead to higher rates of growth and to higher functional complexity for dissipative networks. These are key features for the origin and evolution of life.

## 1. Introduction

### 1.1. The Problem of Evolution

Darwin’s theory of evolution is based on the preferential selection of inheritable variations that improve an organism’s fitness. As Darwin was well-aware, however, natural selection cannot explain the origin of life, and it cannot explain spontaneous self-organization within non-replicating systems. Dissipative structures spontaneously emerge within far-from-equilibrium systems [1], and they are ubiquitous on earth and throughout the cosmos. 

Evolving systems, whether biological or physical, are sustained by the supply and dissipation of free energy. Numerous researchers have searched for fundamental principles to describe the stabilities of non-equilibrium dissipative systems. In 1922, Alfred Lotka proposed that natural selection maximizes the total energy flux through a system, and that “this law of selection becomes also the law of evolution” [2]. The idea was further refined by Howard Odum and Richard Pinkerton as the Maximum Power Principle of ecology [3], and it remains an important concept in the field [4].

Lars Onsager investigated coupled transitions, in which the flow of a component down a potential gradient can push another component up a potential gradient [5]. Heat flow through a thermocouple, for example, drives electrons up a voltage gradient. Onsager’s work documented the power of dissipation to create and maintain non-equilibrium states. 

Ilya Prigogine further investigated the thermodynamics of non-equilibrium states and proposed his minimum entropy production principle. The MinEP principle states that near equilibrium, the rate of entropy production is minimized at a steady state [6]. Near-equilibrium is defined by linearity between flow rates and driving forces. Linearity means that the steady state is the only state consistent with a system’s boundary constraints. With a single allowable state, there is no opportunity for selection or self-organization.

Prigogine’s main interest was the spontaneous self-organization of far-from-equilibrium systems, where nonlinearity allows multiple configurations consistent with a system’s kinetics and boundary constraints [1]. As a system is pushed farther from equilibrium, new dissipative configurations become possible and existing configurations become unstable. The transition of a fluid from conduction to organized convective flow with increased heat input is a familiar and well-documented example of spontaneous self-organization. 

Statistical mechanics describes the stability of a stationary state in terms of statistical probabilities. A system’s stationary state at an instant of time can be represented by a point in a generalized state space, which is spanned by a dissipative system’s state variables. Each point follows a trajectory determined by the system’s kinetics and boundary constraints, and this converges to a stable stationary state, which is represented by an attractor [7]. A point attractor describes a steady state dissipator, a limit cycle attractor describes a stationary cycling system, and a “strange” attractor bounded in state-space represents a stationary chaotic system. 

A far-from-equilibrium system has multiple attractors, each representing a locally stable stationary state with its own basin of attraction. A stationary state is stationary because it is locally stable to small perturbations, but, at the boundaries between basins of attraction, infinitesimal fluctuations can send a system to one of multiple different attractors. Statistical mechanics attributes the sensitivity to fluctuations to deterministic chaos.

An attractor’s basin of attraction is the set of state-space points that deterministically connect to the attractor. E.T. Jaynes described the probability of an attractor by its “caliber”, which he defined as the density of the trajectories that lead to it [8]. If all initial states are equally probable, then the caliber and probability for a stationary attractor is related to the size of its basin of attraction. Deterministic statistical mechanics describes the statistical probability of a dissipative system’s attractor based on an *a priori* assumption of the probability distribution of initial states and the fraction of initial states that deterministically lead to that attractor.

Prigogine and colleagues argued that mechanical instabilities at the boundaries between basins of attraction can amplify quantum fluctuations [9,10], and that this leads to macroscopic randomness. A trajectory might cross multiple boundaries and have multiple opportunities for macroscopic random choices. In addition, as a system’s surroundings changes, the system’s landscape also changes. New attractors can emerge and the boundaries between basins of attraction can shift. Macroscopic randomness at basin boundaries goes beyond the determinism of mechanics, and it provides evolving systems the freedom to explore new possibilities. 

Glansdorff et al. showed that extremal principles can explain the local stability of dissipative states within a basin of attraction [11], but Glansdorff and Prigogine concluded that an extremal principle governing the relative stability of locally stable dissipative states cannot exist [12]. Without a principle of global stability, a dissipative state’s probability is still described by the statistics of its trajectories. Random fluctuations at critical points of instability simply introduce an additional factor in the statistics of a system’s trajectories.

One proposed principle of self-organization that has received considerable attention is the Maximum Entropy Production Principle (MEPP) [13]. The MEPP is based on the idea that “faster is better.” It proposes that if a far-from-equilibrium system has multiple stationary solutions consistent with its physical and boundary constraints, then the one with the highest rate of entropy production is the most stable. This is commonly observed. Conduction and convection can both be consistent with the system’s boundary constraints, but a perturbation will irreversibly trigger a switch from conduction to convection, increasing the rates of dissipation and entropy production. A mixture of methane and oxygen will slowly react to produce water and carbon dioxide, but a spark will irreversibly trigger a switch to hot combustion, increasing the rates of dissipation and entropy production. Simple organic compounds subject to UV-C have been shown to spontaneously self-organize into UV-C pigments, increasing photon absorption and the rates of dissipation and entropy production [14].

The MEPP principle remains controversial, however. In a review article, Garth Paltridge describes how in the 1970s he stumbled upon the entropy production rate as an extremum function that was maximized by atmospheric systems [15]. In 2009, however, he stated that “the concept of maximum entropy production, while not of any immediate practical value in itself (the all-too-normal situation with MEP), was at least a catalyst for investigating something of practical interest in another field.” [16].

An extremum function is key to any comprehensive theory for physical evolution and complexity. The Santa Fe Institute (SFI), an organization dedicated to complexity science, concluded in a workshop held in 1993 that complexity arises in many disparate types of systems, and that there can likely be no simple extremum principle or unified theory of complexity. Twenty years later in a 2014 retrospective [17], David Pines, one of SFI’s cofounders, acknowledged that the dream of a unifying theory of complexity remained elusive. The current state of complexity science basically remains a description and computer simulation of complex systems and their evolution, and it has been debated whether complexity science is really a science at all [18]. 

Physics has moved beyond classical mechanics, but it continues to define the physical state within the Hamiltonian Conceptual Framework (HCF) of mechanics and statistical mechanics [19]. Statistical mechanics regards entropy as a measure of an observer’s ignorance of a system’s actual mechanical microstate, and it does not recognize entropy or irreversibility as fundamental. 

Classical Irreversible Thermodynamics (CIT), as developed by Lars Onsager, Ilya Prigogine and many others, breaks from the HCF and mechanics by accepting empirical irreversibility and random fluctuations as fundamental. Nonlinear CIT explains the local stability of stationary states [10]. It describes the relative probabilities of stationary states based on the initial state distribution and random fluctuations at critical points, and on state-space trajectories determined by nonlinear kinetics [14]. CIT theory successfully describes the evolution of dissipative structures within chemical networks [20] and the abiogenesis of fundamental molecules of life under photochemical potentials [14,21,22,23]. This paper, however, is interested in transcending the description of self-organization to an explanation of self-organization in terms of a global principle of stability. In the author’s opinion, this requires a radically new conceptual framework.

### 1.2. A Thermocontextual World

The thermocontextual interpretation (TCI) provides an alternative to the Hamiltonian Conceptual Framework (HCF) of mechanics [19]. Mechanics describes a system by its microstate, which expresses everything measurable and knowable about a system’s physical state. It defines the microstate by perfect measurement in the absence of thermal noise, with respect to a reference state at absolute zero temperature. Mechanics does not accommodate irreversible change as fundamental. It regards the irreversible flow of time as a matter of perception, and it regards processes as fundamentally deterministic and time symmetrical.

TCI also provides an alternative to CIT. CIT recognizes irreversibility and randomness, but it simultaneously defines a non-isothermal system with respect to multiple reference states. CIT describes a non-isothermal system by partitioning it into separate thermally equilibrated parts, each locally equilibrated with a different-temperature reference state. If the system’s temperature(s) change, the CIT reference state(s) also change. This is the local equilibrium hypothesis.

TCI’s postulates and definitions are summarized in Appendix A, and they provide the conceptual framework for the concepts needed for any discussion of self-organization. TCI is based on a simple premise: the physical state is thermocontextually defined relative to an ambient reference state in equilibrium with the system’s actual surroundings at a positive ambient temperature. This provides a reference for defining entropy, exergy, and irreversible system time as thermocontextual properties of state. The ambient reference state provides a fixed reference from which a process of change can be described. This enables TCI to define dissipation, utilization, and efficiency as well-defined properties of a dissipative process. 

TCI joins a long list of quantum mechanical interpretations [24], but to the author’s knowledge, it is the only one to define the physical state thermocontextually. This has allowed TCI to provide straightforward explanations for long-standing questions without the extreme and untestable metaphysical implications common with existing interpretations [19]. TCI reconciles the determinism and time-symmetry of physical laws with quantum randomness and the thermodynamic arrow of time; it reconciles the superluminal correlation of entangled measurements with relativistic causality; and it resolves the measurement problem of quantum mechanics [19,25]. 

As with any foundation, we need to start at ground level. This paper addresses familiar concepts, but these need to be carefully reassessed from a thermocontextual perspective. TCI defines a dissipative system, not by its state properties, but by its dissipative function (Appendix A). TCI lays the foundation for a new principle of selection for dissipative processes, which is introduced in Section 3.1, and which drives the evolution of open dissipative systems towards higher efficiency of exergy utilization.

## 2. TCI States and Processes

### 2.1. The Thermocontextual State

The Thermocontextual interpretation (TCI) generalizes the definition of a mechanical state [19,25]. As with the Hamiltonian Conceptual Framework, TCI defines a system’s absolute energy with respect to a hypothetical zero-energy reference state at absolute zero temperature. However, TCI then partitions absolute energy, E_abs_, into thermocontextual components, which is given by:(1)$$E_{abs}=E+E_{as}=X+Q+E_{as}.$$

Absolute energy is resolved into system energy, E, and ambient state energy, E_as_. Ambient state energy is the system’s “ground state” energy, and it is defined by equilibrium with the system’s surroundings. It is the energy of the ambient reference state with respect to the hypothetical zero-energy absolute zero reference state. Since the ambient temperature is always positive (Postulate Two), the ambient state energy is always positive.

The system energy is defined relative to the ambient state energy. It is resolved into exergy (X) and entropic energy (Q). Exergy is defined by the system’s potential work capacity on the ambient surroundings in the limit of a quasistatic equilibrium process. TCI further partitions a system’s exergy into the sum of mechanical exergy and thermal exergy. Mechanical exergy is the sum of the system’s particles’ kinetic and non-thermal potential energies. Thermal exergy is the work potential of the system’s thermal energy, q. Thermal exergy is empirically given by:(2)$$dX_{q}=\left(\frac{T-T_{a}}{T}\right)dq,$$
where dq is an increment of heat at temperature T and T_a_ is the ambient temperature. 

Exergy is a generalization of free energy. Whereas exergy and exergy changes are well-defined thermocontextual properties of state, free energy is defined by work that is reversibly measured at the system’s temperature. A reference state for measuring free energy and its changes is not well-defined for a non-isothermal system or for a system with a changing temperature. For a special-case system at fixed and uniform temperature T, however, free energy and exergy are equivalent, with dF = (T/T_a_)dX.

Entropic energy is defined by Q ≡ E − X. A system’s entropic energy and the ambient heat of the surroundings both have zero exergy, and they are freely exchangeable. Entropic energy is related to thermal energy at temperature T by:(3)$$dQ=\left(\frac{T_{a}}{T}\right)dq=T_{a}\left(\frac{C_{V}}{T}\right)dT,$$
where C_V_ is the volumetric heat capacity.

TCI finally defines thermal entropy by:(4)$$S\equiv \frac{Q}{T_{a}}.$$
As described in [19], thermal entropy is a generalization of thermodynamic entropy, which is defined with respect to absolute zero temperature. From (3) and (4), dS = dq/T ≡ dS_td_, and the changes in thermal entropy and thermodynamic entropy are identical. They differ only in their zero-entropy reference state. TCI also defines statistical entropy as a transactional property of a transition between states [25] (Appendix A).

Exergy, entropic energy, and ambient temperature are three independent thermocontextual properties of state, and together, they define a system’s thermocontextual energy state. Other thermocontextual properties include system energy, ambient state energy, absolute energy, and entropy. The energy state is specified by any three independent thermocontextual properties. The energy state for the ambient reference state at T_a_ is uniquely defined by E = X = Q = S = 0 and E_abs_ = E_as_. 

The thermocontextual properties of state are empirically defined by perfect measurement (Figure 1). Perfect measurement involves a thermodynamically reversible and closed transition from an initial state (A) to an ambient reference state (B). Thermodynamic closure means that the system can exchange energy and work with the surroundings, but not mass components. Thermodynamic reversibility means a quasistatic equilibrium process. Heat is isothermally exchanged with the surroundings at the ambient temperature, and there is no dissipation or entropy production.

Perfect measurement reversibly transfers energy to the surroundings as ambient heat q_a_ at the ambient temperature and as utility υ, which we define as the summed transfers of work and thermal exergy (Figure 1). For an open system, utility can also include the internal exergy of exported components. A system’s exergy, entropy, and entropic energy are empirically defined by measurements of the utility and ambient heat reversibly transferred to the surroundings. A system’s exergy is always non-negative, but its initial entropic energy can be negative or positive.

### 2.2. Transitions

A transition describes a spontaneous change in state. Isolated transitions are governed by the Second Law of thermodynamics, which states that the entropy of an isolated system (or system plus surroundings) never declines and is maximized when the isolated system reaches equilibrium. TCI’s Postulate Four (Appendix A) generalizes the Second Law of thermodynamics to apply to all transitions. Postulate Four says that a state of lower exergy is more stable than a state of higher exergy, and that the highest stability state is the equilibrium ambient state, with zero exergy. In contrast to an isolated transition, a non-isolated transition has the option of reducing its exergy, in part, by performing work on the surroundings. The Second Law of thermodynamics is a special case of Postulate Four when applied to an isolated system for which exergy decline is due to internal dissipation alone. 

If a thermodynamically closed transition outputs utility (sum of work plus thermal exergy), it is exergonic (Figure 2a). If the transition is driven by an external supply of utility, it is endergonic. An endergonic transition uses utility from the surroundings to lift a component to a higher exergy (Figure 2b). 

A transition is driven by the dissipation and flow of exergy, not by the flow of entropic energy. To describe the change in exergy during a transition, we start with the fundamental equation for thermodynamic energy change, given by:(5)$$dE=TdS-PdV+\sum \mu_{i}dM_{i}.$$

Equation (5) expresses the conservation of internal energy for systems not subject to external forces or external field changes. It states that the change in internal energy dE equals the sum of inputs of (1) classical entropic energy (TdS) from the direct addition of heat at temperature T and from the entropies of added components, dM_i_; (2) the input of work (represented by −PdV); and (3) the inputs of free energy from added components, dM_i_, with chemical potential μ_i_ (free energy per unit component i). We note that the addition of components’ thermal energy is included in the TdS term.

We can recast Equation (5) within the TCI framework by making the following changes: Replace TdS with T_a_dS=Q (Equation (4)).Replace PdV (PV work output) with generalized utility output, dυ.dE = dX + dQ (Internal energy equals internal exergy plus entropic energy).Replace chemical potential μ with specific exergy, $\bar{X}$ (exergy per unit component).

Inserting these changes into Equation (5) yields:(6)$$dX=-d\upsilon +\sum {\bar{X}}_{i}dM_{i}.\;\;\;\left(\mathrm{For}\;\mathrm{reversible}\;\mathrm{transitions}\right)$$
Irreversible dissipation has no impact on the total energy, and it is ignored by Equation (5), and, therefore, also by (6). Dissipation does affect exergy, however, and TCI’s fundamental equation for internal exergy change includes irreversible dissipation:(7)$$dX={-dQ}_{diss}-d\upsilon +\sum {\bar{X}}_{i}dM_{i}.\;\;\;\left(\mathrm{For}\;\mathrm{all}\;\mathrm{transitions}\right)$$
dQ_diss_ is the production of entropic energy by dissipation and dυ is the output of utility to the surroundings associated with the decline in exergy (Figure 1 and Figure 2a). Equation (7) describes the change in a system’s internal exergy due to internal dissipation and the reversible exchanges of components, utility, and entropic energy with the surroundings. 

An important class of transitions is the thermodynamically closed isentropic transition. Isentropic means that the component’s entropy is fixed, and a fixed entropy means that the isentropic transition is nonstatistical (i.e., deterministic). Isentropic does not mean reversible, however. If irreversibly produced entropy is output as ambient heat, then the transition is isentropic and deterministic, but not reversible.

From the conservation of energy, the decline in exergy during an isentropic entropic transition equals the output of utility and ambient heat. This is expressed by Equation (8):−∆X dM = dυ + dq_a_ (exergonic isentropic transition).(8)
Equation (8) describes the transition’s outputs of dissipated exergy as ambient heat, dq_a_, and of utility, dυ. We can rewrite (8) for an endergonic transition as (9):−dυ = ∆ XdM + dq_a_. (endergonic isentropic transition).(9)
Equation (9) describes the utility dυ supplied to an endergonic isentropic transition to lift an increment of component dM to higher specific exergy and to overcome dissipation (Figure 2b). Dissipation reduces the exergy of outputs for both exergonic and endergonic transitions. 

One of the most familiar transitions is the one-dimensional steady state flow of heat down a temperature gradient (Figure 3a). As described in the figure caption, the transition is exergonic and isentropic. 

If we measure temperatures across the length of steady state conductive heat flow, we find an essentially linear gradient. The steady state rate of heat flow is empirically given by Fourier’s Law, J = k∇T, where k is the thermal conductivity and a constant.

Other phenomenological laws relating flow rates to gradients include Ohm’s law for electrical flows, Fick’s law for chemical diffusion, and chemical kinetics for chemical reaction rates. Phenomenological rate laws describe rates driven by thermodynamic pressure gradients. Examples are given in Table 1, below. In the linear range, flow rates increase linearly with thermodynamic pressure gradients, giving rise to phenomenological rate laws for heat, chemical diffusion, and other processes. Linearity between flow rates and gradients define the near-equilibrium range. The table also shows the differentials in specific exergy for the various components. In the case of electrical and fluid flow, thermodynamic pressure is equal to the exergy gradient, but this is not the case for heat or chemical processes.

In the classical limit, a steady state transition is continuous (Figure 3a). In quantum mechanics and TCI, however, transitions are not infinitely resolvable and elementary transitions are always discrete and finite. An elementary transition is thermocontextually defined as the smallest measurable change in a component’s state. 

We describe an elementary transition as a node that takes a component from one state to another state of measurably different exergy. Internal details of the elementary transition or the component’s state are not measurable, however. Consequently, they are not definable within TCI. TCI defines states and elementary transitions as discrete. Figure 3b illustrates an elementary transition of heat to lower temperature and exergy. 

The decline in exergy across the elementary node is output as the measurable quanta of ambient heat and dissipated energy. Its rate of dissipation is given by:(10)$$\dot{Q}=J\times \Delta \bar{X}.$$
J is the rate of flow for a generic component and Δ$\bar{X}$ is the component’s change in specific exergy. Equation (10) is the general expression for the rate of dissipation across a transition node.

### 2.3. Dissipative Processes

We now proceed to describe an open system’s stationary process of dissipation and its interactions with its surroundings. Figure 4 illustrates the TCI model for a stationary dissipative system, which is sustained by a stable supply of utility and components. The model assumes a stationary environment, but it is nonequilibrium. The system’s surroundings include an ambient background and one or more sources of utility or high-exergy components. A system with stationary utility sources and environment for wastes will converge over time to a stationary process of dissipation. The system is stationary, but it is not static, and the dissipative state is not an actual state, as its components are in constant flux and dissipating exergy. We describe a stationary system’s dissipative process by a dissipative function (Appendix A). 

TCI models the dissipative process as a network of elementary transitions linked by the exchanges of components and energy. An elementary transition has inputs and outputs of components with measurable states, but the internal details are not resolvable or measurable from the ambient surroundings. 

Figure 5a illustrates a near-equilibrium dissipative system with a pair of linked transition nodes, A and B, and components, 1 and 2. Component 1 is the driving component, and at low concentration, component 2 is the driven component. Near equilibrium, flows are linear with generalized thermodynamic pressures. We further assume that the components’ thermodynamic pressures are equal to their exergy gradients. The flow rates, following Onsager [5], are then given by:(11)$$J_{1}=L_{11}\nabla {\bar{X}}_{1}+L_{12}\nabla {\bar{X}}_{2}\;\;and\;\;J_{2}=L_{21}\nabla {\bar{X}}_{1}+L_{22}\nabla {\bar{X}}_{2}$$
where L_ij_ are constants describing the linear contribution of the specific exergy gradient in component j to the flow of component i. Onsager expressed potentials by entropy gradients instead of specific exergy gradients, but they differ from each other only by a constant factor (the negative ambient temperature).

The total power input to the two nodes is shown by the blue line in Figure 5b. Power input is equal to the total dissipation
$\dot{Q}$, where the over-dot indicates its rate. The blue curve shows that dissipation is minimized at steady state for the zero flow of component 2 (
$\dot{\upsilon }$_AB_ = 0) at X_2_ = 2. This is a very special case steady state. For any other fixed value of X_2_, the entropy production rate is not minimized at steady state. We also note that the Onsager’s relations (Equation 10) are based on the equality of thermodynamic pressures and negative exergy gradients, and this is yet another special case, not valid for heat flow, diffusion, or chemical reactions. The rates of dissipation and entropy production are generally not minimized at steady state, even very near equilibrium.

Coupled flows can create nonequilibrium gradients, but if flows and thermodynamic pressures are linearly related, then the steady state is the only dissipative solution. The linear flows in Figure 5 can maintain a nonequilibrium steady state, but there is no opportunity for selection or self-organization. 

Self-organization involves choices, and this requires non-linear dynamics. Figure 6 illustrates a simple non-linear model that can switch between two steady state processes. The Schlögl reaction [26] consists of a single component and two sequential transitions: (1) A→x and (2) x→B. A is the fixed state of input, B is the fixed state of output, and x is an intermediate state that converges to a steady state. Transition 1 describes the transition from state A to x. Its reaction rate is given by k_+_Ax^2^ − k_−_x, where k_+_ and k_−_ are the forward and reverse kinetic rate coefficients. This corresponds to a detailed reaction given by A + 2x→3x. The product x partakes in the reaction, making the system’s dynamics autocatalytic and non-linear. Figure 6a illustrates the nodal network representation of the Schlögl reaction.

The rate of change In x is the difference in the rates of transition from A to x and from x to B. Based on simple kinetic rate theory and Table 1, the rate of change in x as a function of the concentration of x is given by Equation (12), below, and shown by the black curve in Figure 6b.
(12)$$\frac{dx}{dt}=-k_{1+}x^{3}+k_{1-}Ax^{2}-k_{2+}x+k_{2-}B,$$
A, x, and B are the activities (idealized concentrations) of the component states, and the k’s are the forward and reverse reaction rate coefficients for transitions 1 and 2. Equation (12) is a third order polynomial, with three solutions for the steady state condition dx/dt = 0. The steady state at x_2_ is unstable to perturbations. X_2_ is a bifurcation point separating two basins of attraction around the stable steady state dissipative functions x(t) = x_1_ and x(t) = x_3_. 

The red curve in Figure 6b shows the dissipation rate, calculated from Equations (10) and (12), with the input and output of a single component and no exchanges of work. None of the steady states align with an extremum in dissipation (or entropy production) rate.

Dissipative processes are not always steady state. Figure 7 shows the dissipative network for the Brusselator reaction [1]. The Brusselator has inputs of two components: component 1 in state A and component 2 in state B. The Brusselator’s “state” is defined by the concentrations of X and Y, which are represented by the horizontal lines of essentially constant and measurable exergy. They have attached “pods” to accommodate transient storage and fluctuations in component concentrations and flow rates. 

The Brusselator has four reactions, R1 to R4, listed in the figure caption. Transition rates, based on simple reaction rate theory and Table 1, are shown by the arrowhead expressions in the figure. The kinetic rate coefficient for the forward direction is set to unity, and the reverse direction is assumed to be much slower and is set to zero. So, for example, the transition rate for R3 (B + X→D + Y) is simply BX. Reaction 2 is an autocatalytic transition of component 1 from state Y to state X. For reaction R2 (Y + 2X→3X), the transition rate is X^2^Y, making the Brusselator non-linear. Non-linearity allows for the possibility of multiple stationary dissipative solutions for a given set of boundary conditions.

R1, R2, and R4 are exergonic transitions of component 1, represented by nodes 1, 2, and 4. Reaction R3 couples two separate component transitions: X→Y and B→D. Node 3B is another exergonic transition, dissipating component 2 from state B to state D. The component’s exergy is only partially dissipated, however. Some of the exergy performs work on node 3A (wavy arrow). Node 3A is an endergonic transition. It implements utility from 3B to lift component 1 “uphill” from state X to the higher exergy state Y.

Setting the net production rates of X and Y to zero yields the steady state dissipative function $\left|X\left(t\right),Y\left(t\right)\right|$ = $\left|A,B/A\right|$. The nonlinearity of the Brusselator’s kinetics and boundary constraints allows for other solutions. For B > 1 + A^2^, the steady state function’s basin of attraction becomes a point and bifurcation boundary. Any perturbation from the steady state dissipative process sends the system on a transient path that converges to a stationary periodic function. The dissipative function for the periodic solution $\left|X\left(t\right),Y\left(t\right)\right|$ is graphically illustrated in Figure 8 as the limit cycle attractor.

Nicolis and Prigogine [1] introduced the Brusselator as a simplified model for oscillating chemical reactions, such as the Belousov–Zhabotinsky reaction, for which A = 4 × bromate (BrO_3_^−^); B = 3 × malonic acid (CH_2_(CO_2_H)_2_); D = 9 × CO_2_; and E = 4 × bromide (Br^−^) + 6 × H_2_O. A well-stirred system shows fluctuating colors, reflecting oscillating concentrations (Figure 8). In an unstirred system, in which diffusion occurs, a variety of moving spatial patterns can develop [27]. The models of an unstirred system are represented in a pixelated two- or three-dimensional model space. The generalized state space in this case is many- but finite-dimensional, with separate state-space dimensions for the concentrations of X and Y in each pixel of the model space. The system’s trajectory over time traces out the changes in the concentrations of X and Y across the system’s pixelated volume.

To summarize, a system’s dissipative process is defined by its dissipative function, which describes its measurable state properties over space and time. The dissipative function is defined by the trace over time along the system’s attractor in a generalized state space spanning the system’s variable properties (e.g., Figure 8). Relevant properties typically include temperature and the component concentrations of each pixel of the system’s pixelated model space. 

Near equilibrium, linearity between flow rates and thermodynamic pressures leads to a unique time-independent steady state and a single-point attractor. Entropy production and dissipation rates are minimized at steady state only for special cases. 

Far-from-equilibrium, multiple attractors, and stationary dissipative processes can exist, as is consistent with a dissipative system’s kinetic and boundary constraints. The integration of Equation (10) over an attractor’s pixelated model space and a unit interval of time yields the system’s time-averaged dissipation over that time interval. Dissipative processes with higher rates of dissipation (and entropy production) are often more stable, but this is not always the case. In the next section, we propose an extremum principle for the global stability of a dissipative system based on the efficiency of its interactions with the surroundings.

## 3. Evolving Complexity

### 3.1. An Extremum Principle for Dissipative Systems

The Second Law of thermodynamics is an extremum principle for the stability of thermodynamic states. This is illustrated in Figure 9a. The Second Law applies to isolated systems, and it says that the state of maximum entropy is the most stable. TCI generalizes the Second Law with its Postulate Four, which says that the state of minimum exergy is the most stable. This is illustrated in Figure 9b. The transition to a more stable state can be by the irreversible dissipation of exergy or by the thermodynamically reversible output of work on the surroundings. 

TCI proposes an extremum principle for the stability of a dissipative system based on its utilization efficiency. This contrasts with other proposed principles based on the rate of exergy dissipation or entropy production. To define a system’s utilization, we start by resolving exported utility into its external work and exported exergy components:(13)$$\upsilon_{exp}=w+X_{exp}.$$
If exergy is exported to the system’s ambient surroundings, that exergy is lost from the system as external dissipation. Tangible work on the system’s ambient surroundings, on the other hand, creates positive exergy adjacent to and accessible by the system. Equation (14) defines the total dissipation of exergy (q_x_) by the sum of internal dissipation (q_diss_) and external dissipation (X_exp_):(14)$$q_{x}=q_{diss}+X_{exp}.$$
From (13), (14), and the conservation of energy, we have for a single dissipative node:(15)$$\upsilon_{in}=\upsilon_{out}+q_{diss}+\upsilon_{exp}=\upsilon_{out}+q_{x}+w.$$
Equation (15) and Figure 9b describe the utility input (υ_in_ = X_A_) to a single node and its outputs of utility (υ_out_ = X_B_), dissipated exergy, and work. 

If we resolve the single-node transition in Figure 9b into a network of elementary nodes, as schematically shown in Figure 9c, the utility output from one node is input to other nodes within the dissipative network, and we refer to this as internal utility, υ_int_. We then rewrite (15) for an individual node as:(16)$$\upsilon_{in}=w+q_{x}+\upsilon_{int}.$$
Equation (16) describes the transition of a node’s utility input to work on the surroundings, and to internal utility, which is transferred to endergonic nodes within the system. Dissipated exergy is exported to the ambient surroundings.

The system’s ambient surroundings defines a system’s measurable resolution, and it thermocontextually establishes the elementary nodes through which the system’s components transition from one discrete state to another. This allows us to define a system’s utilization (Υ) as the sum of the rates of internal utility transfers plus work exports:(17)$$\Upsilon =\sum \langle {\dot{\upsilon }}_{int,i}\rangle +\sum \langle {\dot{w}}_{i}\rangle .$$
The over-dots are time derivatives, and the angle brackets denote stationary properties averaged over the system’s characteristic unit of time. The characteristic unit of time for a steady state process can be any positive time interval; for a periodic dissipative process, it is one cycle; and for a chaotic system, it is sufficiently long to dampen fluctuations to below the thermocontextually defined limit of resolution. 

We can now define a system’s utilization efficiency (Ξ) by: (18)$$\Xi =\frac{\Upsilon }{{\dot{\upsilon }}_{in}},$$
and we express the maximization of efficiency by Postulate Five.

Postulate Four states that a system spontaneously transitions to a more stable state of lower exergy, and Postulate Five (**MaxEff**) constrains how that transition occurs. For an isolated transition, internal dissipation is the only means of reducing exergy, and this means that an isolated transition maximizes dissipation and entropy production. For an open transition, however, work on the surroundings provides another way to reduce the exergy of output to comply with Postulate Four. MaxEff says that given the opportunity, a system selects exergy utilization over exergy dissipation. 

Many of the fundamental molecules of life are UV-C pigments, which can be derived from simple precursors under UV-C light. In an article in this special volume, Karo Michaelian provides a detailed description of the abiogenesis of Adenine within the framework of Classical Irreversible Thermodynamics [14]. A pigment minimizes the reflection and transmission of incident photons, and within the framework of TCI, this minimizes external dissipation (X_ext_). Reducing the export and external dissipation of exergy makes additional exergy available for internal utility transfers to other dissipative steps within the overall process of dissipation. From (17) and (18), this reflects an increase in efficiency. The abiogenesis of UV-C pigments under a UV-C flux provides a clear illustration of MaxEff.

MaxEff is also an essential basis of measurement. A measurement setup provides a system with an opportunity to actualize a measurement result as it transitions to its measurement reference state (Figure 1). MaxEff ensures that when a measurement is conducted, the transition deterministically records the measurement results rather than dissipating its exergy without recording a measurement. Recording a measurement result is one means of utilizing available exergy. The concept of measurement can be generalized to include recording any exergonic transition, such as sensory inputs.

We have already seen the importance of autocatalysis in dissipative systems. MaxEff helps explain why autocatalysis might spontaneously emerge. A transition from a high-exergy state A to lower exergy state B typically involves an intermediate activated state A^*^. The high specific energy of A^*^ creates a barrier that limits the reaction rate from A to B. This relationship is expressed by the Arrhenius equation [28], relating the kinetic rate constant k to temperature: (19)$$k\propto e^{-\frac{E_{A^{*}}}{RT}}.$$
E is the energy of the activated state A^*^, and R is the universal gas constant. The idea is that at a higher temperature, there is a higher probability of a fluctuation reaching the energy necessary to pass over the activation energy’s barrier, resulting in a higher reaction rate. 

If we rewrite the Arrhenius equation within the TCI framework, we obtain:(20)$$k\propto e^{-\frac{X_{A^{*}}}{Q_{A}}}.$$
RT in (19) is the entropic energy plus ambient state energy of an ideal gas at temperature T. Equation (20) replaces E_A*_ and RT—both defined with respect to absolute zero temperature—with X_A*_ and Q_A_—both defined with respect to the ambient reference state. A higher entropic energy increases the statistical probability of reaching the activated state exergy barrier and increases the reaction rate. Counterintuitively, component A must irreversibly reduce the exergy it had at its input in order to increase its entropic energy and increase its probability of reaching the activation exergy. This results in a higher reaction rate, but lower efficiency. Autocatalysis reduces the activation exergy for the reaction. By reducing the activation exergy, autocatalysis allows a reaction to proceed with a lower entropic energy, a higher utility output, and a higher efficiency. 

MaxEff applies to individual transitions, but its real significance is its application to dissipative networks. MaxEff describes the self-organization of a dissipative network into dissipative structures as a response to increasing its internal transfers of utility. Thermal exergy added to the base of a liquid carries out thermal expansion that sustains convection. Hot combustion heats its fuel supply and maintains a higher reaction rate than cold combustion. In both cases, the more stable process is sustained by internal exergy utilization, and each has higher utilization efficiency than conduction or cold combustion. Convection and hot combustion are two familiar examples of spontaneous transitions to more stable dissipative processes of higher internal utilization and efficiency.

### 3.2. The Two Arrows of Evolution

We can rewrite Equations (17) and (18) as: (21)$$\Xi =\frac{\sum \langle {\dot{w}}_{exp,i}\rangle }{\langle {\dot{\upsilon }}_{in}\rangle }+\frac{\sum \langle {\dot{\upsilon }}_{int,i}\rangle }{\langle {\dot{\upsilon }}_{in}\rangle }=G_{F}+C_{F}.$$

Equation (21) resolves efficiency into two components: the growth factor, G_F_, which corresponds to external efficiency, and the functional complexity factor, C_F_, which corresponds to internal efficiency. A dissipative system can increase its efficiency and stability either by having a positive rate of growth or by increasing its internal utilization and functional complexity.

The first arrow of evolution is growth by expansion or replication. If a dissipative system does work on a component in the surroundings, then that component is no longer part of the zero-exergy ambient surroundings, and a new node is created in a growing network of dissipators. This describes the utilization of ultraviolet photons’ exergy on ambient organic compounds to create high-exergy compounds. 

An existing dissipator can also grow horizontally by expansion or replication. Horizontal growth describes the spread of a fire or a species’ population growth. In the case of a spreading fire, work involves heating ambient fuel in the surroundings to its combustion point, allowing the fire to spread. In the case of a species’ increasing population, work on the environment involves building new biomass from ambient resources. As a system grows horizontally by expansion or replication, its requirement for exergy supply increases in proportion. A dissipative system can grow horizontally up to the carrying capacity of the system’s environment to sustain it.

The second path towards higher efficiency is by increasing internal utilization and functional complexity. The term complexity is commonly associated with the amount of information required to specify a system’s state. Here, we are interested in the complexity of a dissipative system’s functional process, as quantified by its functional complexity, C_F_. The functional complexity is the ratio of internal utilization to utility input, and it is a measurable and well-defined property of a dissipative process. 

For a single pass of exergy and a single endergonic node, functional complexity can approach unity as dissipation approaches zero. However, a dissipative system can increase its functional complexity well beyond unity by reprocessing and recycling exergy via feedback loops or by sustaining a network of exergonic–endergonic pairs. Figure 10 illustrates a simple feedback loop resulting in functional complexity and efficiency of five hundred percent. Feedback loops are ubiquitous within biological systems, from cells to ecosystems, leading to higher functional complexity. Figure 10 could also illustrate a high-exergy system or organism sustained by low-exergy inputs from its environment.

A system can also increase its functional complexity by upward or downward integration. In downward integration, a system incorporates existing nodes that already utilize exported exergy. Downward integration increases a system’s efficiency by internalizing utility transfers from what would otherwise be externally dissipated to the surroundings. An ecosystem can be viewed as a downward integration from plants to herbivores, carnivores, scavengers, and degraders. The ecosystem is sustained by sunlight, and each level sustains the level below. An ecosystem with a stable environment is constantly evolving new niches to recycle nutrients and utilize otherwise wasted exergy [29].

In upward integration, a system incorporates existing exergy sources. Upward integration increases efficiency by converting external utility sources into internal utility transfers. The evolution of eukaryotic cells can be viewed as an upward integration of prokaryotic cells and their incorporation as organelles. 

Growth and increasing functional complexity are two distinct ways that a dissipative system can increase its efficiency and stability.

### 3.3. Oscillations and Synchronicity

Oscillations and synchrony are ubiquitous within biological systems, human institutions, astrophysics, and quantum mechanics [30]. The spontaneous emergence of resonance, commonly observed in mechanical or fluid mechanical systems far from equilibrium, illustrates spontaneous cycling and the arrow of increasing functional complexity. 

The Brusselator illustrates the higher efficiency and stability of an oscillating process over a steady state dissipative process. Its steady state and cycling functions have identical time-averaged rates of dissipation. They differ, however, in their efficiency. For the steady state dissipative process, the concentrations of X and Y are fixed. There is never a measurable transfer of component 1 “uphill” from state X to Y, and endergonic node 3A performs no net work on the system. For the oscillating mode, in contrast, component 1 periodically accumulates in and is released from the X and Y pods. During the phase of the cycle with a net transfer from X to Y, endergonic node 3B performs the internal work of lifting component 1 to higher exergy. The oscillating dissipative process, therefore, has higher efficiency than the steady state mode. MaxEff asserts that the oscillating mode is more stable, in agreement with perturbation analysis. We generalize this conclusion and assert that an oscillating dissipative process is more stable than a steady state dissipative process, other differences being negligible. 

Systems of linked oscillators often synchronize in a process known as entrainment [31]. Figure 11 and Figure 12 illustrate the synchronization of a network of linked oscillators. As detailed in the figure captions, the analysis shows that a network of linked oscillators increases its efficiency when all oscillators synchronize. MaxEff, therefore, predicts that networks of coupled oscillators are stabilized by synchronization, independent of their physical details. 

For a strictly deterministic system, like the model in Figure 12, a dissipative process’s stability is related to the size of its basin of attraction. Many randomly assigned starting points settle into full synchronization like that shown in the figure. For others, however, the system deterministically settles into a variety of other attractors, including partially synchronized dissipative functions, in which subsets of oscillators are synchronized but the subsets are phase-shifted from each other. They have efficiencies that are intermediate between full synchrony and asynchrony. With sufficient time and intrinsically random transitions [25], a system would generally settle into its most stable process of synchronization and maximum efficiency. However, an external jolt is sometimes required, such as the need for a defibrillator to restore the synchronization of pacemaker cells. 

MaxEff provides a general principle that explains the spontaneous emergence of oscillations and synchronization in terms of a general principle, independent of a system’s specific dynamics. Figure 12 illustrates the power of MaxEff over statistical mechanics by being able to efficiently assess global stability without a statistical analysis over many different starting points.

### 3.4. Whirlpools Disprove the MEPP

We commonly observe whirlpools, indicating that they can be more stable than radial flow of water directly towards a drain. A whirlpool provides an important counterexample to proposals that processes are stabilized by maximizing the rate of dissipation or entropy production. The Maximum Entropy Production Principle (MEPP) and related proposals have had success in a number of areas, but they are not universally applicable [15,16]. 

MEPP equivalent to maximizing the rates of net utility supply and dissipation. The centrifugal force of a whirlpool’s circulation lowers the water level and pressure over the drain, and this actually reduces the rate of water discharge. A stationary whirlpool, therefore, has lower rates of water and exergy input, a lower rate of dissipation, and a lower rate of entropy production. A whirlpool’s stability results from its higher rate of internal utilization and higher functional complexity. 

To model a whirlpool’s functional complexity, we define a pixelated model space, which is represented in Figure 13. It represents a 50 cm diameter cylindrical container of water partitioned into concentric shells radiating outward from a central drain. Water in each shell is modeled with uniform pressure and kinetic energy. 

Water in the outermost shell is maintained at a constant 20 cm depth. Drainage is assumed to be turbulent and proportional to the square root of the pressure of water over the drain, taken as the height of the water column in the central core. At each interface, a component of water transitions to a zone of lower pressure, higher velocity, and higher kinetic energy. This applies to both the radial flow and whirlpool models. 

Figure 14 shows a detail of the network model for the transition between two zones in the whirlpool model. The surface water contour for the whirlpool is determined by conserving angular momentum and balancing hydrostatic and centrifugal forces. 

For radial flow, the surface water contour is determined by energy conservation, with increasing kinetic energy towards the drain offset by lower potential energy and water column height. Viscous dissipation outside of the central core is negligible for radial flow and is ignored.

Figure 15 shows profiles for the radial flow and whirlpool models. In the case of radial flow (solid lines), the profiles show that the drain-ward decline in water level and the increase in kinetic energy are imperceptible at the plotted scale. In the case of whirlpools (dashed lines), the figure shows a dramatic decline in water pressure and height near the drain. This profile represents the whirlpool’s shape. There is a correspondingly large increase in water velocity and kinetic energy as water approaches the drain. Figure 15 clearly shows that the increase in velocity and kinetic energy near the drain is much greater for the whirlpool than for radial flow.

Table 2 summarizes the results of the steady state radial and whirlpool models. The table shows that the rates of water input and output are higher for the radial flow, but the internal work rate is 4000 times higher, and the functional complexity is 5000 times higher for the whirlpool. According to MaxEff, the whirlpool should be more stable than radial flow, despite its lower rates of net power, dissipation, and entropy production. The common emergence of whirlpools in draining water empirically documents the relative stability of whirlpools over radial flow, as predicted by MaxEff. The stability of the whirlpool provides an important counterexample to the idea that “faster is better.” It falsifies the Maximum Entropy Production Principle, which asserts that the rate of exergy dissipation and entropy production always tends to be maximized.

## 4. Summary and Discussion

Physics defines the physical state relative to a theoretical reference state at absolute zero temperature. Thermocontextual interpretation (TCI), in contrast, defines the physical state relative to a reference state in equilibrium with the system’s actual surroundings. This allows TCI to resolve a system’s total energy into exergy, which can perform useful work on the positive-temperature ambient surroundings, and entropic energy, which is the randomized energy of ambient heat. Exergy is an easily measurable quantity, but it is a thermocontextual property and it is not recognized by physics or thermodynamics. Statistical mechanics and thermodynamics describe the Second Law only in terms of entropy. Without thermocontextual properties of state, they cannot recognize the dissipation or utilization of exergy; all they can see is the production of entropy. 

TCI goes beyond statistical mechanics and Classical Irreversible Thermodynamics: first, by establishing exergy as a physical property of state and the measure of its instability, and then by generalizing the Second Law of thermodynamics. The Second Law applies to isolated systems and states that irreversible dissipation minimizes exergy and maximizes entropy. TCI’s Postulate Four extends the Second Law’s application to non-isolated systems. It maintains that a system irreversibly minimizes its exergy, but it can do so either by dissipating its exergy or utilizing it. Utilization describes the action of a dissipative system, either of sustaining other dissipative systems within a network of linked dissipators or of performing work on the system’s surroundings to expand its reach. The Maximum Efficiency Principle (MaxEff) goes one step further and states that a dissipative system tends to minimize the dissipation of exergy and maximize efficiency to the extent that is possible.

MaxEff sheds light on two of the most perplexing questions: the origin and definition of life. Natural selection can explain the evolution of life, but it cannot account for its prebiotic origin. By preferentially selecting dissipative processes of higher efficiency, MaxEff drives open systems to expand their dissipative networks and to increase their functional complexity. Autocatalytic loops for recycling exergy (e.g., Figure 10) lead to higher efficiency through increased functional complexity, and self-replicating units lead to higher efficiency through positive growth (21). Once self-replicating networks emerge, natural selection can then act to propagate beneficial variations and drive the evolution of life.

One of the defining characteristics of life is self-replication according to a “blueprint” that is subject to modification and selection. A system’s blueprint for self-replication is, in essence, a form of autocatalysis. It directs a source of exergy to autocatalyze a system’s replication from ambient resources. As described in Section 3.1, autocatalysis is one mechanism to achieve higher efficiency. 

Another defining characteristic of life is the ability to interact with and learn from the environment. Even simple bacteria can learn correlations from their environmental interactions and can adapt their future behavior accordingly [32]. The process of learning starts with the work of making and recording measurements of environmental interactions. As described in Section 3.1, recording measurements is another mechanism for achieving higher efficiency. Learning also involves training to synchronize a system’s functions to the environment, further reducing dissipative losses and increasing efficiency (e.g., Figure 12). Vitaly Vanchurin, in fact, takes learning as a starting point for a theory of evolution [33]. Self-replication and learning from the environment are two defining characteristics of life, and both can be explained by MaxEff.

Once life is established, it continues to evolve through an interplay between competition and cooperation. Competition drives growth, and cooperation drives an increase in functional complexity. These are the two basic paths for evolving systems. Competition dominates when resources are plentiful and when the dissipative cost of competition is small relative to the gain in utilization from increased resource supply and growth. Competition enables a species to expand its resource base and to achieve higher efficiency through growth.

Cooperation dominates when resources are inelastic. When a rainforest’s canopy completely covers the forest, its solar resource base is maximized, and it is inelastic to further gains. Over its fifty-million-year period of relative environmental stability, the Amazon rainforest has nevertheless continued to evolve, by developing increasingly complex webs of interacting dissipators acting together to recycle components and exergy [34]. Ecological nutrient cycling [29] involves the repeated utilization and renewal of the nutrients’ specific exergy. From Equation (21), nutrient recycling increases the system’s internal utilization and functional complexity factor. Given a stable environment, evolution is open-ended, and a dissipative system can increase its functional complexity with no theoretical limit. 

Any model is an approximation of an actual system that is valid within its domain of applicability. Quantum mechanics does not invalidate classical mechanics within its domain of its applicability, and TCI does not invalidate the Hamiltonian Conceptual Framework (HCF) or Classical Irreversible Thermodynamics (CIT) within their domains of applicability. TCI does, however, extend the range of applicability to describe thermocontextual properties for systems as they exist with respect to surroundings at a positive ambient temperature. It explains intrinsic randomness by instantiation (Figure 1 and Appendix A) during the reversible export of entropic energy to the surroundings, and it provides a fixed reference from which changes across time can be defined. TCI is a generalization of the HCF and CIT, and where their valid applications overlap, there is strict correspondence in their conclusions. TCI also applies to changes in ambient surroundings. A decrease in the ambient state energy leads to refinement, random instantiation, and spontaneous symmetry breaking [19,25]. Spontaneous symmetry breaking during expansion and ambient cooling of the early universe may resolve some of the outstanding questions of cosmic evolution, such as the baryogenesis problem [35].

By recognizing thermocontextual properties of state and MaxEff, TCI establishes a firm foundation for the description and analysis of self-organization. A foundation is an essential step, but a systematic description of self-organization requires building upon that foundation. It requires a formalized mathematical framework for constructing nonlinear transition operators, which must include a system’s kinetic and thermocontextual boundary constraints. The stationary functions consistent with a system’s transition operator can then determine the dissipative processes, and MaxEff would provide an efficient way to determine their relative efficiencies and stabilities without the numerous measurements or simulations that are often needed for a statistical analysis of probabilities. 

## Figures and Tables

**Figure 1 entropy-25-00229-f001:**
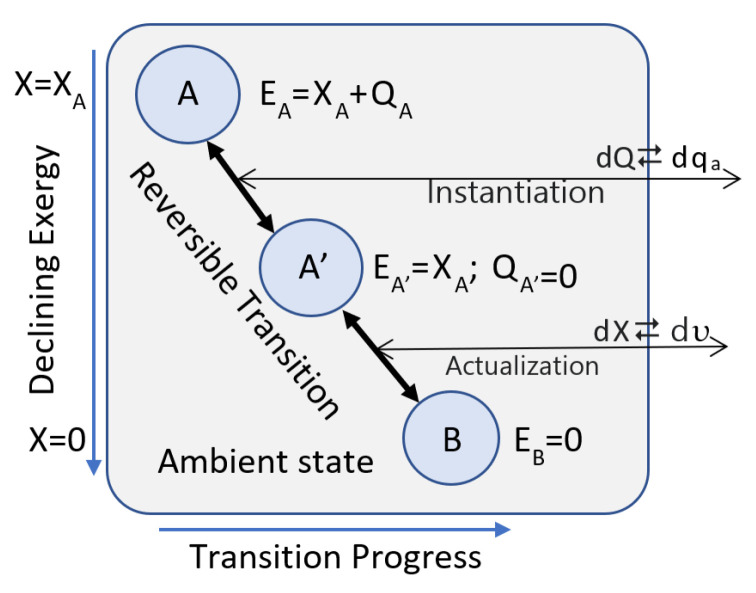
Perfect measurement is a reversible closed-system transition from an initial State A to ambient State B. The measurement of a positive-entropy system is resolved into two steps. The first step for a positive entropy state involves derandomization by the reversible transfer of entropic energy to the surroundings. This sets the statistical entropy σ_A′B_ (Appendix A) [25] to zero and randomly instantiates a zero-entropy microstate for state A′. The second step is the actualization of the measurement results. This involves the reversible and deterministic transfer of state A′, with exergy X_A_ to ambient state B, and the export of exergy to actualize a measurement result.

**Figure 2 entropy-25-00229-f002:**
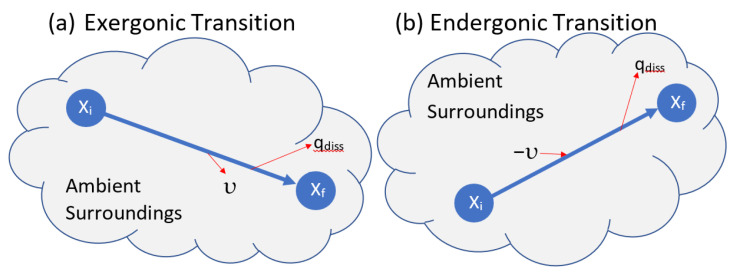
A component transitions as a thermodynamically closed system, exchanging energy and work but no material components with the surroundings. (**a**) In an exergonic transition, a component transitions to lower exergy and outputs utility (υ). (**b**) Utility added to an endergonic transition lifts a component to higher exergy. Dissipation reduces the efficiency of both exergonic and endergonic transitions.

**Figure 3 entropy-25-00229-f003:**
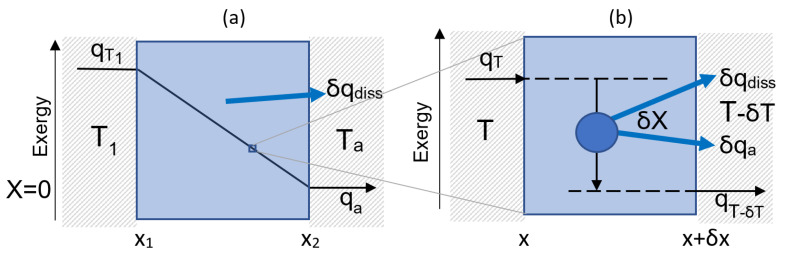
(**a**): Continuous heat flow in the classical limit. Input energy q_T_ is resolved into thermal exergy X_q_ and entropic energy Q. The input of entropic energy is output as ambient heat, q_a_. Thermal exergy is dissipated and is also output as ambient heat, but as q_diss_. Produced entropy is exported, and the transition is, therefore, isentropic. (**b**): Discontinuous heat flow through an elementary transition node.

**Figure 4 entropy-25-00229-f004:**
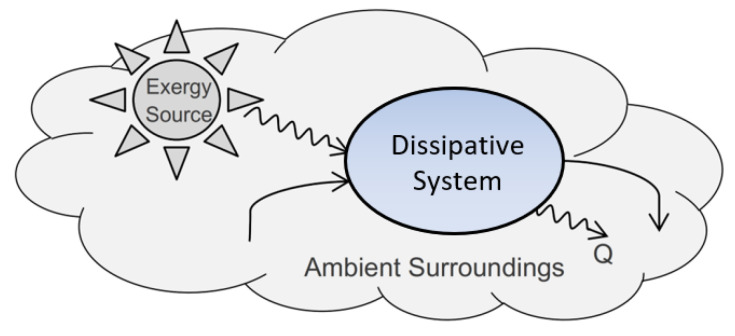
Dissipative system model. The system’s nonequilibrium surroundings include utility source(s)—either directly (e.g., sunlight) or indirectly—by high-exergy components. Other components in the surroundings (e.g., water, air, detritus) are also freely available to the system for processing and discharge. A stationary system is defined by a function describing its stationary dissipative process, with equal time-averaged inputs and outputs of materials and total energy.

**Figure 5 entropy-25-00229-f005:**
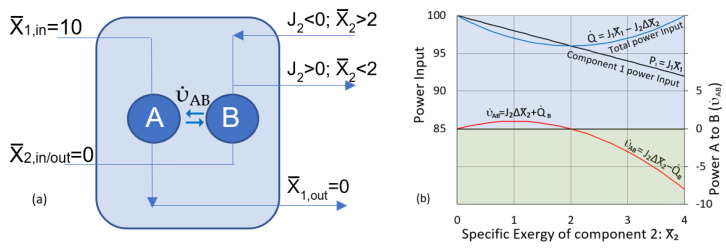
Coupled transitions. The flows of components 1 and 2 are based on Equation (11) with L_11_ = L_22_ = 1 and L_12_ = L_21_ = 0.2. (**a**) For X_2_ < 2, exergonic node A outputs power to endergonic node B (
$\dot{\upsilon }$_AB_ > 0), which lifts component 2 to higher exergy. For X_2_ > 2, the flow of component 2 is reversed, and node B becomes exergonic, outputting power to node A. (**b**) Graphs of power input for component 1 (black line) and total power input (blue line). Total power input equals total dissipation
$\dot{Q}$. The figure also shows the rate of utility transfers between nodes ($\dot{\upsilon }$_AB_) (red line).

**Figure 6 entropy-25-00229-f006:**
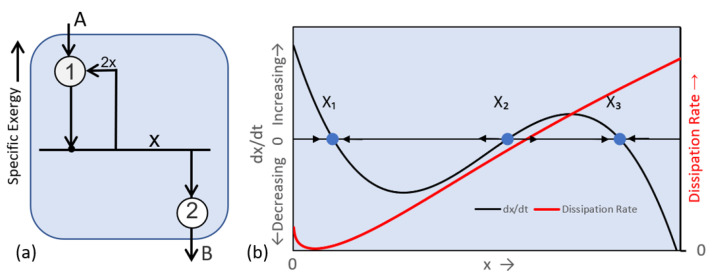
(**a**) Nodal network diagram for the Schlögl Reaction. Transition 1 is an autocatalytic reaction A + 2x→3x. Transition 2 is a simple transition x→B. (**b**) The black curve shows the rate of change in the concentration of x versus concentration. The Schlögl reaction’s three steady states, at x_1_, x_2_, and x_3_, are defined by dx/dt = 0. Perturbation analysis shows that x_1_ and x_3_ are stable to perturbations. The red curve shows the system’s rate of dissipation as a function of the concentration of x.

**Figure 7 entropy-25-00229-f007:**
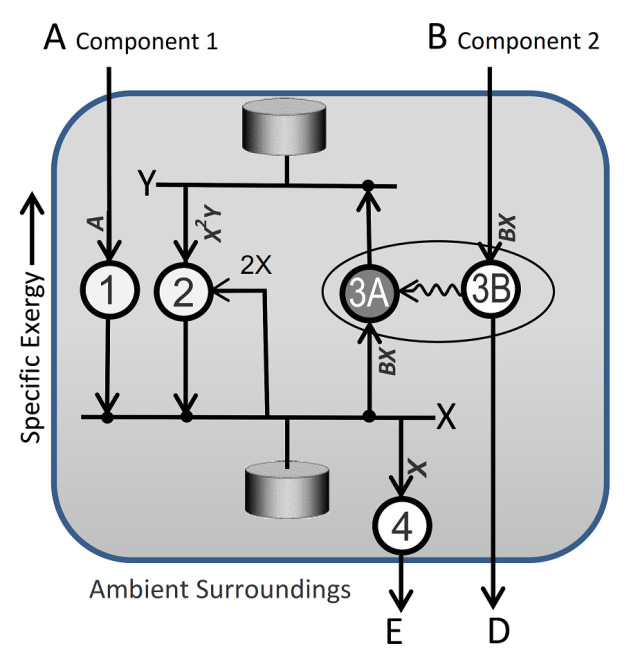
The Brusselator network model. Letters refer to the components’ states. The overall reaction is A + B→E + D. States higher on the diagram have higher specific exergy. The Brusselator comprises four reaction steps: R1: External source A → X; R2: Y + 2X → 3X; R3: External source B + X → Y + D (discharge); and R4: X → E (discharge). The arrowhead expressions describe reaction rates.

**Figure 8 entropy-25-00229-f008:**
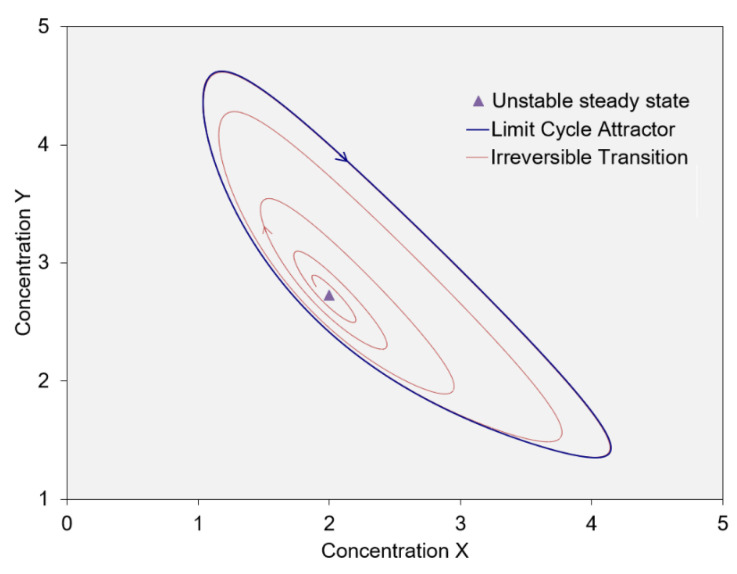
The Brusselator’s stationary dissipative functions in state space, spanned by its state variables. The arrows show the direction of change in the stationary oscillating state over time within a single well-stirred volume.

**Figure 9 entropy-25-00229-f009:**
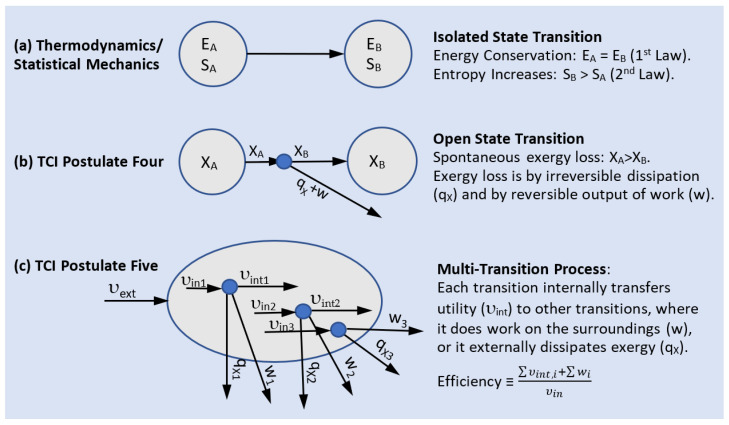
Internal and external utility. (**a**) Thermodynamics describes an isolated transition by the conservation of energy and production of entropy. (**b**) Postulate Four extends the Second Law to open transitions, in which exergy decline can be by dissipation to q_X_ and by external work w. (**c**) For a network of dissipators, an elementary node’s utility output is resolved into external utility (work) and internal utility, which is input to other nodes within the network.

**Figure 10 entropy-25-00229-f010:**
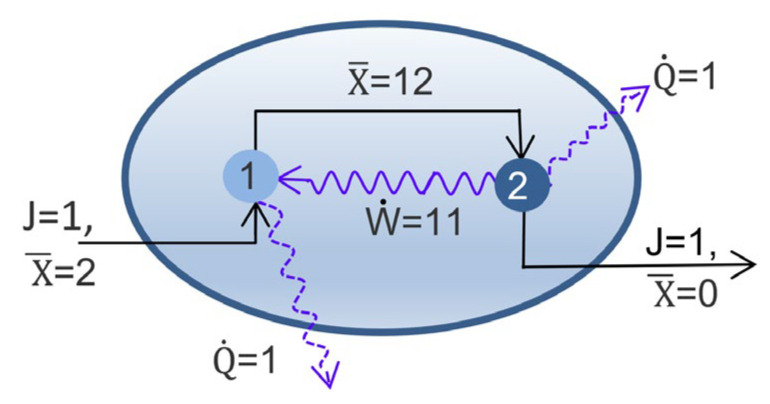
Simple feedback loop. A component flows through a system (straight vectors) at a rate (J) of one mass unit/s. The component’s specific exergy ($\bar{X}$) equals two units at input and zero at output. The rates of net exergy input (J × $\bar{X}$) and total dissipation rate ($\dot{Q}$) equal two energy units/s. Mass and energy inputs and outputs are balanced. Exergonic node 2 takes the component with 12 units of specific exergy, dissipates one unit, and performs work on endergonic node 1 (wavy vector) at a rate of 11 units/s. Endergonic node 1 applies ten units of work to lift the component’s specific exergy from 2 units to 12 units, and in the process, it dissipates one unit of work. The system’s rate of internal work on the component equals 10 units. With 2 units of exergy input, its functional complexity C_F_ and efficiency equals five.

**Figure 11 entropy-25-00229-f011:**
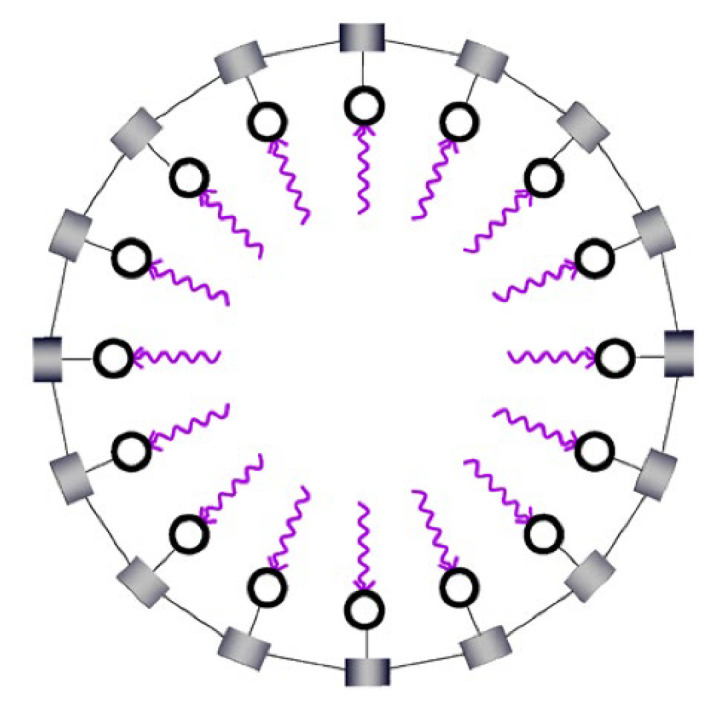
Coupled oscillators. The figure shows sixteen oscillators linked in a circle. All oscillators have identical unit rates of utility input (wavy arrows) into nodes, which pump an ambient component (not shown) into the pods. When the concentration in a pod reaches a critical value, the component is released, and the cycle resumes. Coupling between adjacent pods allows components to leak from one pod to another. The leak rate equals the difference in concentrations, multiplied by a coupling coefficient.

**Figure 12 entropy-25-00229-f012:**
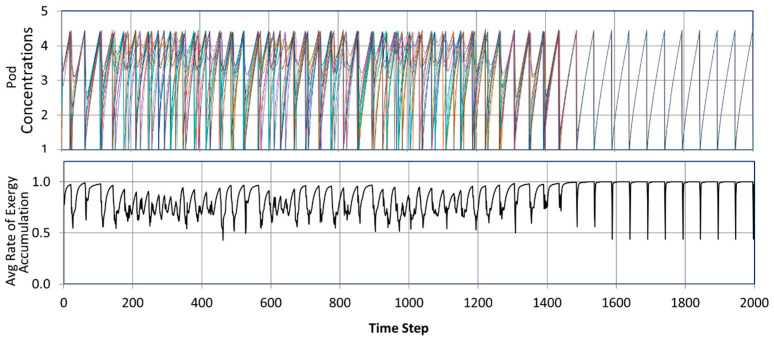
Synchronization of coupled oscillators. Top—Pod Concentrations: Oscillators are randomly assigned periods between 49.5 and 50.5 time units. They start with randomly assigned concentrations. After about 1500 time steps, the oscillators synchronize. Bottom—Average Rate of Exergy Accumulation: Each oscillator has a unit rate of work input, which is used to pump component into its pod. When oscillators are not synchronized, some exergy is lost to diffusive leakage between adjacent oscillators. When all oscillators synchronize after 1500 time steps, concentrations are equal, there is no diffusive loss, and the pods’ rate of exergy accumulation is maximized and equal to the work input (except at pod discharge). The rate of internal work is equal to the time-averaged accumulation of exergy by the pods, and this is maximized by synchronization. Synchronization, therefore, increases the efficiency of the system’s dissipative process.

**Figure 13 entropy-25-00229-f013:**
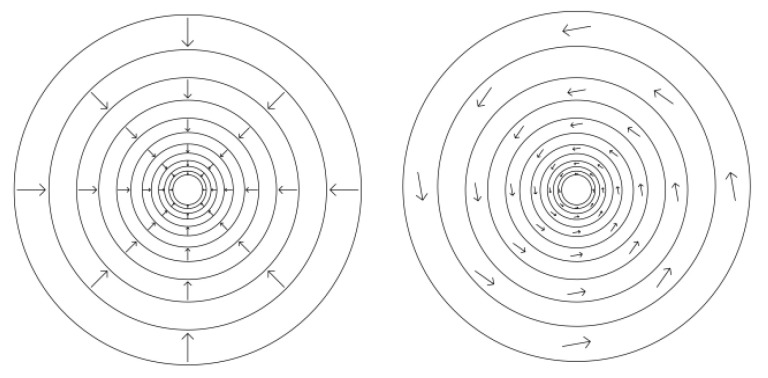
Models for radial flow and whirlpool. Left: radial flow. Right: whirlpool. Each concentric shell is a single zone with uniform pressure (water elevation) and fluid speed. Arrows illustrate fluid flow directions only. Speed is constant within each zone but the radial speed increases inwards in both cases due to the incompressibility of water. In addition, the conservation of angular momentum requires that the rotational velocity for the whirlpool is inversely proportional to the radial distance and increases inwards.

**Figure 14 entropy-25-00229-f014:**
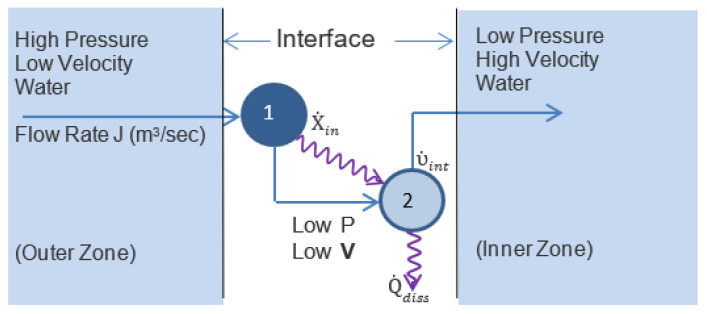
Paired nodes at whirlpool zonal interface. As water flows across a zonal interface, it undergoes both a decline in pressure and an increase in velocity and kinetic energy. An elementary node represents only a single transition, so each interface has two nodes. The first node is exergonic. It transfers exergy at a rate of ${\dot{X}}_{in}=J\Delta P$ to endergonic node 2. Node 2 uses this exergy for the internal work of accelerating the water, which is given by ${\dot{\upsilon }}_{int}$ = ${\dot{X}}_{in}-{\dot{Q}}_{diss}=\text{½}\rho {\Delta V}^{2}J$, where ${\dot{Q}}_{diss}$ is the rate of dissipation by node 2. The internal work for the system is summed over all interfaces.

**Figure 15 entropy-25-00229-f015:**
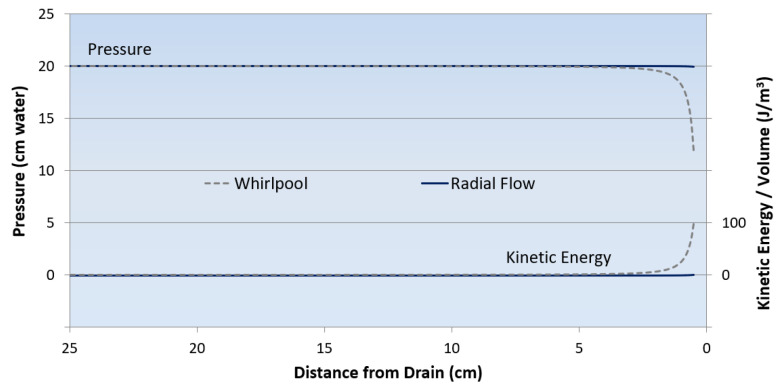
Pressure and kinetic energy profiles for whirlpool and radial flow. Fluid velocity and kinetic energy increase towards the drain for both the whirlpool and radial flow. The solid lines show an imperceptible drop in the pressure and a similarly imperceptible acceleration of water for radial flow. The dashed lines show an 8.1 cm (40%) pressure drop and a sharp acceleration of water for the whirlpool near its vortex. The lower pressure at the drain for the whirlpool corresponds to lower rates of water discharge, dissipation, and entropy production compared to radial flow.

**Table 1 entropy-25-00229-t001:** Linear rate laws and exergy differentials.

Process	Component	Thermodynamic Pressure (Generalized Concentration)	Phenomenological Rate Law	Specific Exergy (dX)
Conductive heat flow, J	Unit heat	Temperature, T	J = −k∇T	X = q(T − T_a_)/T; dX=qT_a_dT/T^2^
Chemical diffusion, J	Unit mass	Chemical activity, ɑ. ɑ ∝ C for dilute concentration, C.	J = −D∇C	X = RT_a_*ln*(ɑ) dX=RT_a_dɑ/ɑ
Electrical flow, I	Unit charge	Voltage, V	I = −σ∇V (σ electrical conductance)	dX = dV
Chemical reaction	Σν_i_R_i_⇌ Σν_j_P_j_ ^(1)^	$\Theta \equiv \frac{\prod a_{P_{i}}^{\nu_{i}}}{\prod a_{R_{j}}^{\nu_{j}}}\equiv \frac{\Theta_{P}}{\Theta_{R}}$	$\frac{d\zeta }{dt}=k_{+}\Theta_{R}-k_{-}\Theta_{P}$ ^(2)^	$\begin{array}{c} d\bar{X}\equiv {d\Theta }_{\bar{X}}\equiv \\ \frac{\prod {\bar{X}}_{P_{i}}^{\nu_{i}\zeta }}{\prod {\bar{X}}_{R_{j}}^{\nu_{j}\left(1-\zeta \right)}}d\zeta \end{array}$ ^(3)^
Laminar flow, J	Fluid	Pressure, P	J = −K∇P ^(4)^	dX = dP

^(1)^ Reactants R_i_ transition to products P_j_ as a closed system. ν_i_ are stoichiometric coefficients. ^(2)^ zeta ζ is a reaction progress variable (0→1) and k_+_ and k_−_ are forward and reverse kinetic rate constants. ^(3)^ Specific exergy $\bar{X}$ = RT_a_ln(ɑ), where activity for the ambient reference state is set to unity. ^(4)^ Hydraulic conductivity K is constant for Newtonian laminar flow, but at higher flow rates, turbulent flow becomes non-linear with pressure gradient.

**Table 2 entropy-25-00229-t002:** Comparison of functional complexity for radial flow and whirlpool models.

Steady State Flow Rate J (m^3^/s)	${\dot{X}}_{\mathrm{in}}$ ρgh_o_J (J/s)	${\dot{X}}_{\mathrm{out}}$ ρJ^3^/A_drain_^2^ (J/s)	Net Power ${\dot{X}}_{\mathrm{in}}-{\dot{X}}_{\mathrm{out}}$ (J/s)	${\dot{\upsilon }}_{\mathrm{int}}$ Σ${\dot{\upsilon }}_{\mathrm{int},i}$ (J/s)	C_F_ ${\dot{\upsilon }}_{\mathrm{int}}$${\dot{X}}_{\mathrm{in}}$
**Whirlpool** 3.00 × 10^−5^	0.029	0.0022	0.027	2.9 × 10^−3^	0.1
**Radial flow** 3.88 ×10 ^−5^	0.038	0.0047	0.033	7.4 × 10^−7^	1.9 × 10^−5^

$\dot{X}$_out_ = Kinetic exergy of water exiting a 1 cm diameter drain with area A_drain_. ρ = Fluid density (1000 kg/m^3^). h_o_ = Water depth at perimeter (20 cm). ${\dot{W}}_{i}=\frac{1}{2}\rho \Delta V_{i}^{2}$ is the increase in kinetic energy per volume of water at interface i (Figure 14). $\dot{W}$
_int_ = Internal work of accelerating water from the perimeter zone to the core zone.

## Data Availability

No new data were created.

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
