# Peer review of "Dissipation + Utilization = Self-Organization"

_entropy, 2023, doi:10.3390/e25020229_

Round 1

Reviewer 1 Report

This article is a serious and meaningful work, which is important not only for thermodynamics as such, but also for a number of related interdisciplinary areas. The thermocontextual interpretation (TCI) of open dissipative systems proposed in the article, including the fourth postulate of TCI and the principle of maximum efficiency (MaxEff), seems to be quite reasonable.

The emphasis made in the article on the thermodynamic description of open dissipative systems is of great interest for studying the problem of the origin of life. This is especially important in connection with the fact that oscillatory processes are rarely taken into account in studies of prebiotic chemical evolution. The author's theses on the re-distribution of exergy in an open dissipative system due to both its internal dissipation and utilization; about the high stability of such systems due to oscillatory processes; about the potential for the emergence of synchronization in biological processes due to pre-biological the synchronization of a network of linked oscillators are directly related to this problem. In fact, they limitlessly expand the possibilities for a prebiotic system to choose the optimal chemical pathways in the course of initial evolution.

The work certainly deserves publication. But there are a number of questions to the author, to which I would like to receive answers.

1. Thermodynamic aspects of the origin of life can be described both in the traditional terminology "free energy - entropy" and in the terminology "exergy - entropy" substantiated by the author. What does the author see as the advantage of the latter for the analysis of this problem?

2. In Section 4, the author touches briefly on biological processes and the origin of life. Is it possible to characterize the transition from an inanimate chemical system to a living biological system in the terminology "exergy - entropy" (after all, we understand that these are fundamentally different types of natural systems, including from the point of view of thermodynamics)? Maybe through some balances of exergy and entropy?

3. What is the difference between the terms negative entropy and exergy, is it possible to put an equal sign between them?

Reviewer 2 Report

After reading certain key sentences in the manuscript I have spent two days for the reading and understanding of the TCI as written in the Author's first paper on the subject published in Entropy 2021 (here ref 15). I admit that I am highly impressed. I like the way the author insists on working as realistically as possible (without dwelling into the definition of realism too much). For example, defining a whole scientific framework such as thermodynamics on absolute zero temperature and completely isolated systems, that both do not exist, cannot be approached from the outside—"There's Nothing Outside" the Universe (Parmenides)— or is unachievable by definition, are not his thing (mine neither). So, the Author set out and formulated with high precision, logically consistent and scholarly clearly a thermocontextual framework based on systems bathing in environments having an ambient positive temperature. He discovered a lot of new things that are in the realm of physics, both classical and quantum physics. Most intriguingly, the technique of separating the notion of time into different qualities, viz. irreversible (external clock) 'reference time', irreversible (thermodynamic) 'system time' and reversible (imaginary) 'mechanical time', and mathematically formulating all these components (using complex space), shows the Author's genius. What I missed in this first paper is a comparison of his definition of a "perfect measurement" with the definition of a measurement in the classical interpretation of physics, and in quantum physics (but without TCI). But never mind, it's never too late.

I went on reading the first and second versions of his paper in Preprints 2022 on the TCI of the double-split experiment. I noticed an inconsistency between the two version, as from equation 8 onwards (statistical entropy versus Gibbs entropy), saw that the first version was much more detailed than the second one (MaxEnt), so I finished reading the second version and keep the first version for later, to figure out what and why there are two versions. I thought it was time to read first the manuscript.

I was excited to see that "Crecraft goes chemical"! I should add right away: "... but not enough to my gusto". I added all comments, minor and major (15 comments) into the attached manuscript file. Here I should like to point out what I missed most.

A) Although nominally not missing, I need quite early in the text a clear discussion on what is the difference between free energy and exergy. This is because formally, mathematically, at least at first sight, exergy replaces free energy but, yes, free energy refers to the free energy available in a thermodynamic context, whereas exergy refers to "free energy" that can be exported as work as measured by an ambient environment, that is, in the TCI. It would be great to see explicit examples (at least one) shat show(s) the clear difference between the two, which brings me back to

B) the aforementioned perfect measurement. I am missing here the notion of heat capacity, since volumetric heat capacity is the measurement in a thermodynamic context which ties internal energy with thermodynamic entropy and, from there, Gibbs free energy. (And what's heat capacity in statistical thermodynamics?)

C) Generally, I think an introductory glossary (maybe in a à la mode box?) where, in addition to the notions of the numerous definitions, also postulates, appear right away, but also those that do not, e.g. reversible, dissipative function, thermodynamic pressure, exergonic, endergonic, Brusselator, utility, utilisation, efficacy, growth, complexification, and so on. This is just a suggestion.

D) Last but not least, I must mention here that, if I hadn't all this respect of your mind and way of thinking, I would have added "Reconsider after major revision", but I don't. You should decide for yourself whether you want to go deeper in this paper or maybe only in some later article, depending on the preliminary results that you would/could obtain from following my comment that I added in the legend of figure 7. Let me explain. I am a chemist. I feel like we chemists are heavily indebted to the scientific community, and the society on the whole, to bridge the gap between physics and biology, where you are reaching out from the realm of physics. When I read about the Brusselator network model, I am fascinated by the simplicity of the model that already gives way to multiple solutions among which are oscillations and chaotic or point attractors, but it also makes me sigh because I cannot connect with it to make it living. The Belouzov-Zhabotinsky reaction with all its wonderful variants does show the most important concept, yes, but just this, doesn't it. We don't have a living world based on bromate burning malonic acid or some other simple organic compound just because this reaction network can be an oscillating one when pushed into it. I think that your insight in MaxEff, that is, that when work can be done then better it'd be done efficiently, namely, periodically, oscillating, is absolutely clue to the question of the origin of life (lines 564-576). I am and always have been convinced about the fact that life is about working efficiently. Primitive complex chemical systems started out transforming exergy into work but, alas!, not very efficiently first. Much of what is the origin of life, and life per se, is about linking in to create coupled oscillators. To work more efficiently, to waste less. As you state, for this to happen we need nonlinear kinetics to unfold (by fuelling properly with high-energy components). There is another reaction that includes DNA and/or RNA fragments that autocatalytically generate nonlinear kinetic relationships. In addition to this, these molecules have been at the heart of making systems replicate. You mention self-replication in lines 714-716 but you do not develop it formally, why not? It should work. (I thought: You shouldn't be in a hurry, hopefully!) It would help the scientific community enormously, if you could show what kind of conditions in the TCI framework could lead to oscillations of the production of nucleic acids. The model that I propose in my comment is based on solid experimental work (not mine) but still is a model, a 'wet' model. In my estimation it should be simple enough to be reasonably resolved in the TCI framework and complicated enough to make us organic chemists connect with it.

If it is possible to explain a priori a way (how in the TCI framework) to make hereditary information molecules reset their mechanical time to zero each time they have (been) replicated and divided into to new 'daughter' sub-systems, thus, have grown and perhaps with thermodynamic system time passing complexified themselves and, most importantly, their surroundings, well, then we can start doing this oscillation job in the lab until it's alive and kicking! Cellular life is Nature's coupled oscillation per se. Starting life through periodicity of key events that become more efficient by MaxEff.

My wife, not a scientist, immediately understood your MaxEff principle. In fact, she said, it would be more efficient to use a car's fuel when, instead of filling her up completely and carry that weight to drive around until it's empty, one would add every morning the amount of fuel needed for the day.  Thermodynamically she's right, and thermoconceptually you are.

I would love Harrison Crecraft to read a study on the equivalence of temperature and time published at https://jsystchem.springeropen.com/articles/10.1186/1759-2208-1-11 which is based on empirical work published earlier at https://jsystchem.springeropen.com/articles/10.1186/1759-2208-1-2. If you would agree and would like to go deeper, you are welcome to contact the main author of these articles (me), in order to be furnished with the (correctly formatted) mathematical appendix of the earlier paper. Please, don't feel obliged to do this, only if you're curious about it, only once you find the mechanical time and exergy, and don't be afraid of telling me the truth of your thoughts without hesitation, I can take it. I published at the time what went through my mind seven years before publication (the empirical part was 10 years of intense work) and, as you know very well, system time is irreversible but mechanical time isn't! One of the things that I haven't quite resolved is the distinction between 'reversible and reversed' and 'reversible but not reversed'.

https://www.icbms.fr/en/membre/109-pierre-strazewski.html

Round 2

Reviewer 1 Report

The manuscript has been improved and should be published.

Reviewer 2 Report

This second version has become easier to understand in some of the critical notions of the TCI, such as the extremum principle, the effects of autocatalysis and the "learning" interaction between the system and its immediate environment, by discussing examples and comparisons that are accompanied with more bibliographic references and Figure 9 is better illustrated. Also, the appendix containing the postulates and definitions is more complete and easier to read.

Minor suggestions:

"Classical irreversible thermodynamics (CIT)" appears for the first time on line 125 and should be explained there, rather than much later on line 856.

Line 133: In this paper, however, we are interested in going beyond the description of self-organization, ...